# Heavy Metal(loid) Accumulation in the Ovarian Tissue of Free-Ranging Queens and Bitches Inhabiting Highly Polluted Urban Environments

**DOI:** 10.3390/ani13040650

**Published:** 2023-02-13

**Authors:** Giovanni Forte, Federica Ariu, Beatrice Bocca, Giuliana Solinas, Giovanni Giuseppe Leoni, Andrea Podda, Roberto Madeddu, Luisa Bogliolo

**Affiliations:** 1Department of Environment and Health, Italian National Institute for Health, 00161 Rome, Italy; 2Department of Veterinary Medicine, University of Sassari, 07100 Sassari, Italy; 3Department of Biomedical Sciences, University of Sassari, 07100 Sassari, Italy

**Keywords:** environmental pollution, metals, ovary, cats, dogs

## Abstract

**Simple Summary:**

Animals are sensitive indicators of environmental pollution and human exposure to environmental contaminants, especially heavy metal(loid)s which have adverse effects at very low concentrations. Cats and dogs meet the demands that define a good sentinel species as they share the human environment and their lifespan is long enough to show the effects of exposure over time. We studied the level of essential and non-essential heavy metal(loid)s in the ovaries of free-ranging queens and bitches of different ages inhabiting highly polluted and non-polluted urban areas of the island of Sardinia, Italy. Our main results revealed an increase in non-essential metals (i.e., cadmium and lead) in the ovarian tissue of animals living in the contaminated sites and an age-related bioaccumulation of these metals. These findings indicated that the accumulation of heavy metal(loid)s in feline and canine ovaries reflected the quality of the inhabited environment and provided new insights for the use of companion animals as sentinels to study female reproductive toxicity of environmental pollution by these chemicals in humans and animals.

**Abstract:**

There is strong scientific evidence that exposure to environmental contaminants, such as heavy metal(loid)s (HMs), can impair female reproductive function. Pets, such as cats and dogs, who share the same habitat as humans, may be particularly useful sentinel models for detecting HMs in the ovary. In the present study, we compared the concentration of essential (Ems; Cu, Fe, Mn, Se, and Zn) and non-essential metal(loid)s (NEMs; Al, As, Cd, and Pb) in the ovarian tissues of free-ranging queens and bitches of different ages living in industrialized/highly polluted (south group) and non-polluted (north group) urban areas of the island of Sardinia, Italy. The results showed that both EMs and NEMs were present at detectable concentrations in feline and canine ovaries and their levels varied according to geographical areas and animal age. Among the EMs, Cu was found elevated in older queens and bitches inhabiting the southern area. Cadmium and lead were higher in feline and canine ovaries of older animals from the south compared to those living in the north. In addition, Cd and Pb concentrations increased in individuals of both species living in the south. These findings showed new perspectives for the use of pets as early warning sentinels of environmental pollution by HMs and for the risk of human exposure within a “One Health” approach. Pets may help to study the link between exposure to metals and female reproductive disturbances in mammals.

## 1. Introduction

The impact of increasing levels of environmental pollution on female reproductive health and fertility is an alarming problem that has aroused growing interest over recent years [1,2,3]. Among the numerous environmental pollutants, heavy metal(loid)s represent potential hazards for human and animal female reproductive function due to the high emission rates [4].

The term heavy metal(loid)s (HMs) has been widely used in the scientific literature and commonly refers to a group of elements that have a high atomic weight and a density greater than 4 g/cm^3^ [5,6]. Some HMs such as copper (Cu), iron (Fe), manganese (Mn), zinc (Zn), and selenium (Se) are considered essential metal(loid)s (EMs), as they mediate vital biological functions when in normal levels, but may become toxic at higher concentrations and adversely affect many physiological and biochemical processes [7]. Other HMs such as aluminum (Al), arsenic (As), cadmium (Cd), and lead (Pb) are not considered to have any vital or beneficial effect on organisms (non-essential metal(loid)s, NEMs) and may be harmful at all concentrations [8].

Anthropogenic activities including mining and smelting operations, industrial, agricultural, and pharmaceutical productions have dramatically increased HMs levels in terrestrial and aquatic ecosystems, leading to changes in environmental exposure. In addition, ongoing climate changes are enhancing the mobilization of naturally occurring HMs with a consequent augmented level in the air and the water [9,10].

Heavy metal(loid) contamination is a serious One Health issue. Being non-biodegradable and persistent pollutants, HMs accumulate in the soil, water, and the food chain and may have detrimental impacts on the ecosystem and consequently on the health of living organisms [11]. These chemicals enter the animal and humans mainly through respiration from air, contaminated food intake, or absorption through the skin, accumulating over time in organs, including the ovaries [12,13].

The female gonad is a heterogeneous organ which plays a key role in female reproductive function. It contains the reserve of primordial follicles and it is the site of oocyte maturation and sex steroid hormone production [14,15]. The accumulation of NEMs in the ovary and changes in EMs content can pose a threat to the reproductive potential of a female because HMs can damage the reserve of ovarian follicles, can interfere with the processes of folliculogenesis and oogenesis, and with the ability of the ovary to synthesize and metabolize sex steroid hormones [16].

Although HMs can impair fertility in both sexes [17,18,19], females are more affected because the gamete pool within the ovaries is fixed at birth and non-renewable. Moreover, in comparison to spermatogenesis, the process of producing fertilizable oocytes is longer and more complex.

Epidemiological and experimental data that were obtained from human and animal research converge to indicate that HMs have an adverse effect on ovarian function and female fertility, even at relatively low-level exposures [19]. Harmful consequence to HM exposure can be observed in females throughout their reproductive life, from prenatal development to puberty and maturity [4,20,21]. The potential health disorders caused by HMs toxicity include interference with the hypothalamic-pituitary-ovarian axis, steroidogenic dysfunction, impairment of ovulation, pregnancy complications, and fetal abnormalities [1,22]. Additionally, it has been demonstrated that HMs enhance oxidative stress, which has detrimental effects on the quality of female gametes and the development of embryos [23,24,25]. In humans, the exposure to HMs has been related to reproductive disorders such as premature ovarian insufficiency and polycystic ovarian syndrome [18,26]. 

Although HM levels have been extensively investigated in different organs such as liver, kidney, and skeletal muscles in mammalian species [13,27], only few studies have been devoted to the analysis of their accumulation in the ovarian tissue [23,28,29]. Most of the studies on humans were performed by the assessment of metal concentrations in the follicular fluid (FF) and granulosa cells [28,30,31] of women undergoing in vitro fertilization cycles. However, some of them exhibited low or undetectable metals levels in FF or follicle–follicle variability [28], thus making the results difficult to interpret and limiting the estimation of the real exposure risk [32]. Sentinel animals has been considered for monitoring environmental HM exposure [33,34]. Post-slaughter ovarian tissues from farm animals have been used to detect the accumulation and localization of HMs [23,29]. Martino et al. [23] measured the levels of metal elements in ovarian tissues of juvenile and adult sheep; the localization of Fe, Zn, and Se was assessed by X-ray fluorescence imaging in relation to anatomical structures in bovine ovaries [29]. However, data that were obtained from farm animal models are mainly representative of the exposure in rural environments but do not give an indication of exposure levels in urban agglomerations. Therefore, more information is needed in order to evaluate HMs accumulation in the female gonads.

Companion animals may be more appropriate sentinel models when estimating the quality of environmental conditions in human surroundings [34]. Cats and dogs share the human habitat more than any other species, their diet is linked to the waste of human nutrition, and they survive a relatively long time compared to farm animals, thus allowing assessment of a longer exposure metal accumulation. Furthermore, ovarian tissues deriving from routine surgical sterilizations of queens and bitches provide an easily accessible and cost effective material to detect metals levels in ovarian tissues.

In this context, we conducted an exploratory study assessing the concentrations of EMs (Cu, Fe, Mn, Se, and Zn) as well as NEMs (Al, As, Cd, and Pb) in ovarian tissues of free-ranging queens and bitches inhabiting industrialized/highly polluted and non-polluted urban areas of the island of Sardinia (Italy). Among EMs, Cu, Fe, Mn, Se, and Zn play important roles in female reproductive function [35,36] and changes in the levels of these elements has been associated with infertility [37,38,39]. The correlation between the concentration of the selected HMs and the age of individuals was also investigated.

## 2. Materials and Methods

### 2.1. Sample Collection

Ovaries were harvested from free-ranging individuals of domestic queens (*Felis catus*) and domestic bitches (*Canis lupus familiaris*) undergoing gonadectomy for population management reasons at the Veterinary Teaching Hospital of Sassari University (Sassari, Italy) and at the Veterinary Clinic of Portoscuso (Cagliari, Italy). The surgeries were performed independently to the accomplishment of this study. To avoid metal contamination, all the procedures were performed using surgical a stainless steel knife and pliers. The ovaries were isolated from the encapsulating ovarian bursa, placed in plastic tubes with very low metal release, and stored at −20 °C until the analysis.

Ovaries were grouped according to the living areas of animals: (i) The south group included 26 free-ranging queens and 21 free-ranging bitches resident in the urban area of Portoscuso-Portovesme, a pullulated area known for environmental heavy metals contamination via industrial processes [40], situated in the south-western cost of Sardinia island (Italy); (ii) The north group consisted of 14 free-ranging queens and 24 free-ranging bitches inhabiting the urban area of Bosa, a small town on the west coast of north-central Sardinia, with a very poor industrialization.

The ovaries from each group were divided into sub-groups according to the age of the animals (queens: ≤1 year and 2–5 years; bitches: ≤3 years and 4–7 years).

### 2.2. Metal Content Analysis

Ovarian samples were examined to detect concentrations of EMs and NEMs that had cumulated in animals from the two parts of Sardinia indicated above.

Approximately 250 mg of the tissues was dried at 105 °C overnight (the mean water content was 72%). Further, all the samples were weighed in 15 mL polystyrene tubes, added with 1 mL of ultrapure concentrated HNO_3_ (VWR, Leuven, Belgium) and digested on a heat block at 80 °C until complete dissolution (ModBlock, CPI International, Santa Rosa, CA, USA) and then diluted with ultrapure deionized water. Milli Q water of the same volume was processed in parallel as a control to determine metal contamination from plastics.

Analyses were performed by sector field inductively coupled plasma mass spectrometry (SF-ICP-MS), using a Thermo Scientific Element 2 model (Bremen, Germany). The low resolution mode (LR, m/Δm = 300) was used for the determination of ^114^Cd and ^208^Pb; the medium resolution mode (MR, m/Δm = 4000) for the determination of ^27^Al, ^65^Cu, ^56^Fe, ^55^Mn, and ^64^Zn; and the high resolution mode (HR, m/Δm = 10,000) for the determination of ^75^As and ^82^Se. An additional calibration method was used to quantify the elements and, as internal standards, the concentration of 1 ng/mL in the analytical solutions of ^115^In in LR and HR and ^69^Ga in MR were used [41,42].

With reference to quality control issues, accuracy, precision, and limit of detection (LoD) and quantification (LoQ) were evaluated. To this end, two certified reference materials (CRMs) were used. The first one was the pig kidney-based ERM-BB186 (Institute for Reference Materials and Measurements, Geel, Belgium) and the second was the Seronorm Trace Elements Whole Blood L-2 (SERO, Billingstadt, Norway). The accuracy ranged from 95% (Fe in pig kidney ERM-BB186) to 112% (As in Seronorm Whole Blood L-2). In addition, the precision that was calculated on replicated measurements of the CRMs was less than 7% in pig kidney ERM-BB186 and less than 5% in the Seronorm Whole Blood L-2. Moreover, LoD and LoQ were calculated based on 3- and 10-times the standard deviation of replicated measurements of blanks, respectively. LoDs and LoQs were, respectively, 0.01 and 0.035 ng/mL for As, Cd, Mn, and Pb; 0.1 and 0.35 ng/mL for Cu and Se; 0.25 and 0.85 ng/mL for Al and Zn; and 1 and 3.5 ng/mL for Fe. Finally, blanks that were calculated in plastic tubes were at the level of LoD for all the investigated elements.

### 2.3. Statistical Analysis

The results were expressed as the mean ± standard error of the mean (SEM) on a dry-weight basis. Since data were not normally distributed differences in element concentrations in both female cats and dogs, correlations between the elements were analyzed using non-parametric tests (U Mann–Whitney and Spearman’s r) setting a significance level of *p* < 0.05. The IBM SPSS Statistics 28 software was the used statistical package.

In order to compare the total content of EMs in ovarian tissue of animals living in the two different geographical areas and by age, the essential metal index (EMI) was calculated as: EMI = (Cm_1_ × Cm_2_ × ⋯ × Cm_n_)^1/n^ where Cm_n_ is the concentration of metal in the sample. Similarly, to compare the accumulation of NEMs in the ovaries, the non-essential metal index (NEMI) was calculated [43].

## 3. Results

### 3.1. Metal Content in the Ovarian Tissue of Queens

The mean concentrations and the standard errors that were determined for the HMs that were analyzed in the ovarian tissue of free-ranging queens in relation to living area are shown in Table 1. With regards to the EMs levels, the Fe concentration was significantly lower in the south than in the north group (*p* ≤ 0.01). No significant differences in NEMs levels were observed between samples of the two groups.

Taken the age-dependent HMs concentration into consideration (Table 2), within the south group, a higher content of Cd (*p* < 0.01) was detected in the samples from queens 2–5 years old compared to those from animals ≤ 1 year. Instead, no difference was recorded in the samples of animals that were living in the north area for the different age groups. Differences in HMs levels were observed when animals of the same age but living in north and south were compared (Table 2). With regards to animals with ≤1 year, Cu increased and Fe decreased in the south group compared to the north one (*p* < 0.05). In the samples of animals 2–5 years old, Fe was lower in the south compared to the north group (*p* < 0.01); instead, Cd and Pb were higher (*p* < 0.05) in the ovarian tissue from individuals living in the south.

Regarding EMI (Figure 1a) and NEMI (Figure 1b) values, the EMI did not differ between the south (11,362 ± 483 ng/g) and north (11,858 ± 677 ng/g) groups and according to the age of the animals (south: ≤1 year, 11,753 ± 736 ng/g; 2–5 years, 10,734 ± 409 ng/g. north: ≤1 year, 10,430 ± 930 ng/g; 2–5 years, 12,930 ± 803 ng/g).

Within the south group, the older cats had a value of NEMI that was significantly higher with respect to the younger group (128 ± 27 ng/g vs. 73.1 ± 14.0 ng/g; *p* < 0.05). The NEMI in cats of 2–5 years was increased in the south group compared to the north one (128 ± 27 ng/g vs. 63.0 ± 15.7 ng/g; *p* < 0.05).

### 3.2. Metal Content in the Ovarian Tissue of Bitches

The mean concentrations and the standard errors of EMs and NEMs that were detected in the ovarian tissues of free-ranging bitches in relation to living area and age are shown in Table 3 and Table 4, respectively. No significant differences in the EMs levels were observed between the samples of the two groups. In the south group, the Cd (*p* < 0.05) and Pb (*p* < 0.01) concentrations of the total population were significantly increased compared to the north one.

The levels of Cd (*p* < 0.01) and Pb (*p* < 0.05) in the ovarian tissues of individuals 4–7 years old living in the south area were higher when compared to those of animals ≤ 3 years. With regards to the younger animals, Cu was found to be significantly elevated in the south group compared to the north one (*p* < 0.05); whereas, in animals 4–7 years old, Cd and Pb were higher (*p* < 0.01) in the south than in the north group (Table 4).

Figure 2 reports data for EMI (a) and NEMI (b) calculated for canine ovarian tissue. No statistical differences for EMI were found when the two geographical areas (south, 9926 ± 832 ng/g; north, 8651 ± 720 ng/g) and the different ages (south: ≤3 years, 10,589 ± 1218 ng/g; 4–7 years, 7530 ± 930 ng/g; north: ≤3 years, 9042 ± 1073 ng/g; 4–7 years, 9772 ± 1028 ng/g) of bitches were compared. With reference to NEMI values, in the south group this index was significantly increased in the samples from the older animals compared to the younger group (37.1 ± 4.4 ng/g vs. 25.0 ± 2.7 ng/g; *p* < 0.05). Moreover, in animals 4–7 years, the NEMI increased in the south group compared to the north one (37.1 ± 4.4 ng/g vs. 27.7 ± 4.9 ng/g; *p* < 0.05).

## 4. Discussion

The use of dogs and cats as sentinels for monitoring naturally occurring environmental and human health hazards has been previously explored [33,34]. Companion animals are ideal surveillance tools to humans because they share the same environment and they are likely to have similar sensitivity towards specific metabolic and clinical responses to specific toxicants [44].

With particular reference to HMs, several studies have addressed the levels of EMs and NEMs in the blood [45], liver, kidney [27,46,47,48,49,50], and in hair [43] of dogs and cats inhabiting rural and urban environments, evaluating the relationship with different animal variables such as diet, sex, age, and pathological lesions on metal accumulation.

Very limited information has been reported regarding the presence of HMs in canine and feline reproductive systems. In particular, Rzymski et al. [43] explored the content of HMs in uterine and testicular tissues of free-ranging and household cats proving that higher concentrations of EMs as well as increased levels of NEMs were present in free-ranging cats.

To the best of our knowledge our study reports, for the first time, the concentrations of HMs in the ovaries of companion animals. We examined the levels of selected EMs and NEMs in the ovarian tissue of free-ranging cats and dogs living in a pullulated urban area such as the industrial estate of Portoscuso-Portovesme in the south of Sardinia Island, and in control urban areas in north Sardinia with low anthropogenic impact. The industrial area of Portoscuso-Portovesme houses the biggest non-ferrous metallurgical complexes with facilities for the processing of aluminum and other metals in the Mediterranean region [51]. Many pollutants may be emitted by aluminum smelting complexes, including toxic metals [52] which can have adverse effects on the health of workers and domestic animals [53,54,55].

The results of our exploratory study showed that both EMs and NEMs were present at detectable concentrations in feline and canine ovaries and that HMs levels varied according to geographical areas and animal ages. In particular, an increased accumulation of HMs has been detected in the ovarian tissue of animals inhabiting the urban area of Portoscuso-Portovesme, thus providing additional evidence of intensive environmental pollution in this area. The dogs and cats that were studied were all free-ranging animals that usually consume garbage, possibly made up of food scraps from meals made at home. Therefore, the exposure to HMs in these animals may be indicative of human exposure.

In order to evaluate the general status of metal elements in ovarian tissue, we used simple indexes (EMI and NEMI) that were previously introduced by Rzymski et al. [43]. Our study found no significant variation in the EMI in relation to habitat and age in both animal species. However, of the five essential metals, Cu levels were significantly increased in younger queens and bitches living in the south, possibly as a result of increased exposure due to Cu environmental pollution coming from industrial activities and/or natural sources. Copper is necessary in maintaining the functioning of living organisms, and plays an important role in female reproduction [38]. However, it is also potentially toxic above optimal levels. Relevant data that were obtained from in vivo and in vitro experiments that were performed on humans and other mammals indicated that Cu may interfere with female reproductive functions [39].

More importantly, the main research findings showed that feline and canine ovaries were a district where the bioaccumulation of NEMI was age-related and reflected the quality of the inhabited environment. Specifically, NEMI value increased in the ovarian tissues of queens and bitches living in the polluted south area with a higher concentration of Cd and Pb in the older animals compared to the younger ones. These findings suggest that women residing in the study area may be at risk for NEMs accumulation in their ovaries over time. In a previous study, we demonstrated that Cd accumulation in ovine ovaries is age-related where significantly higher concentrations of Cd were detected in the ovaries of adult ewes (7.89–24.39 ng/g) compared with those of lambs (0–2.7 ng/g). The levels of Cd in the ovaries of adult ewes were of a lower magnitude compared to those we detected in the ovarian tissue of cats inhabiting the south area.

Our results also showed a clear and evident difference in Cd and Pb concentrations between the two species. Ovarian tissues of queens accumulated much higher levels of these two metals even at younger ages compared to those of bitches. The free-ranging cats could feed not only on human waste found in urban areas but also on their natural prey, therefore, their intake of toxic metals was likely to be higher than that of free-ranging dogs.

Cadmium and Lead are listed by the World Health Organization (WHO) as toxicants of major public health concern (https://www.who.int/news-room/photo-story/photo-story-detail/10-chemicals-of-public-health-concern accessed on 1 June 2022).

There is growing evidence that these metals may significantly alter the functioning of the mammalian ovarian function [19]. Cadmium and lead can act as endocrine disruptors in the ovary interfering with hormone signaling pathways, [12,30]. Animal experimentation revealed that Cd and Pb exposure induced significant changes in the female gonads including decreased follicular growth, increased number of atretic follicles, impairment of ovulation, and degeneration of the corpus luteum [19]. Furthermore, in vitro exposure to Cd and Pb negatively affected oocyte maturation and subsequent fertilization and embryo development in different animal species [25,50,56,57,58,59,60]. Cadmium can be highly toxic even at low doses, as we demonstrated in a previous study where exposure of ovine cumulus oocytes complexes to nanomolar concentration of Cd during in vitro maturation impaired oocyte fertilization and caused oxidative damage [23].

The results of our pilot study provide preliminary but new information for the use of pets as sentinel animals in the One Health approach. Biomonitoring HMs accumulation in canine and feline ovarian tissue may serve as an early warning for environmental pollution by HMs in unknown polluted areas or to monitor already polluted sites. Sentinel pets may be used as surveillance for the risk of exposure to humans that may share the same environment and to monitor HMs bioaccumulation in the ovarian tissue. Finally, they could potentially provide useful information regarding the association between accumulation of HMs in ovarian tissue and female reproductive disorders and may give guidance to develop protective strategies to counteract HMs reproductive toxicity.

Our findings provide direction to continue the study considering a wider number of animals as well as to explore the relationships between HMs accumulation in ovaries and lifestyle, environmental, and dietary variables. Further studies are obviously warranted to investigate the effects and underlying mechanisms of HMs accumulation on the morphology and physiology of the female gonads in order to improve knowledge on the link between HMs exposure and female reproductive disturbances in mammals.

## 5. Conclusions

In summary, the results of the present study demonstrated that easy access to the ovaries removed during sterilization of queens and bitches enabled HMs to be assessed directly in the gonad. Feline and canine ovaries contained detectable levels of essential and non-essential metals which varied in relation to environmental pollution. Cadmium and Pb were found to be most prevalent in the gonads of animals inhabiting urban pullulated areas and accumulated with age. Our findings may provide new insights for the use of pets as early warning sentinels of environmental pollution by HMs and for the risk of human exposure within a One Health perspective.

## Figures and Tables

**Figure 1 animals-13-00650-f001:**
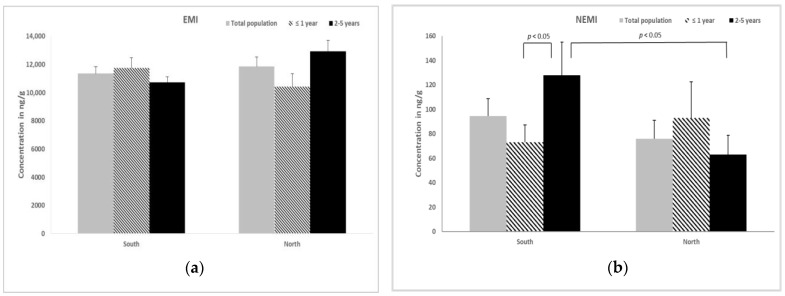
The mean ± SEM of (**a**) the essential metal index (EMI) and (**b**) the non-essential metal index (NEMI) in the ovarian tissues of free-ranging queens of different groups.

**Figure 2 animals-13-00650-f002:**
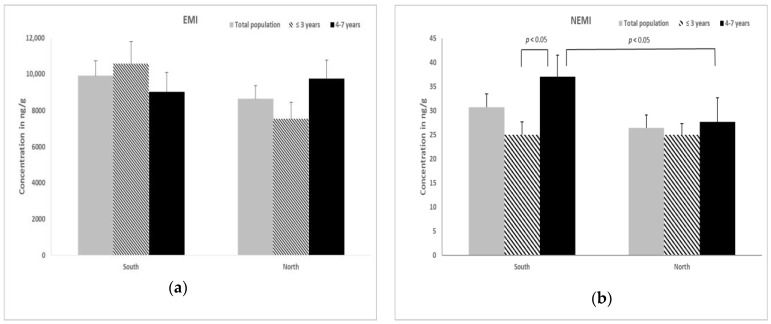
The mean ± SEM of (**a**) the essential metal index (EMI) and of (**b**) the non-essential metal index (NEMI) in ovarian tissues of free ranging bitches of different groups.

**Table 1 animals-13-00650-t001:** The levels of essential and non-essential metals (mean ± SEM) in ovarian tissues of free-ranging queens inhabiting two different areas of Sardinia.

Metal (ng/g)	Geographical Area	
	South	North
Essential		
Cu	6220 ± 722	5044 ± 757
Fe	271,574 ± 16,518 ^a^	401,336 ± 30,907 ^a^
Mn	1239 ± 170	1352 ± 152
Zn	66,503 ± 3784	66,107 ± 3192
Se	1617 ± 59	1562 ± 75
Non-essential		
Al	614 ± 147	887 ± 287
As	139 ± 39	107 ± 61
Cd	52.2 ± 14.0	26.4 ± 5.5
Pb	81.4 ± 16.6	51.1 ± 17.9

^a^ statistically different at *p* ≤ 0.01.

**Table 2 animals-13-00650-t002:** The levels of essential and non-essential metals (mean ± SEM) in ovarian tissues of selected age groups of free-ranging queens.

Metal (ng/g)	Geographical Area			
	South		North	
	Age			
	≤1 Year (*n* = 16)	2–5 Years (*n* = 10)	≤1 Year (*n* = 6)	2–5 Years (*n* = 8)
Essential				
Cu	6785 ± 1064 ^a^	5171 ± 741	3472 ± 576 ^a^	6224 ± 1104
Fe	272,821 ± 21,783 ^b^	269,578 ± 26,543 ^e^	346,739 ± 12,031 ^b^	442,284 ± 49,679 ^e^
Mn	1275 ± 102	1181 ± 88	1173 ± 322	1444 ± 122
Zn	67,830 ± 5004	64,379 ± 5985	68,087 ± 6483	64,621 ± 3141
Se	1652 ± 87	1559 ± 65	817 ± 259	940 ± 481
Non-essential				
Al	767 ± 226	384 ± 127	817 ± 259	940 ± 481
As	114 ± 36	180 ± 87	192 ± 141	43.1 ± 12.2
Cd	21.6 ± 3.1 ^c^	101 ± 31 ^c,f^	27.7 ± 7.7	25.4 ± 8.2 ^f^
Pb	36.3 ± 6.3 ^d^	149 ± 30 ^d,g^	54.6 ± 31.8	48.6 ± 22.4 ^g^

^a,b,f,g^ statistically different at *p* ≤ 0.05; ^c,d,e^ statistically different at *p* ≤ 0.01.

**Table 3 animals-13-00650-t003:** The levels of essential and non-essential metals (mean ± SEM) in ovarian tissues of free-ranging bitches inhabiting two different areas of Sardinia.

Metal (ng/g)	Geographical Area	
	South	North
Essential		
Cu	3228 ± 334	2469 ± 250
Fe	326,658 ± 35,738	277,120 ± 30,525
Mn	1318 ± 180	1142 ± 165
Zn	59,784 ± 8494	58,913 ± 8218
Se	1497 ± 115	1357 ± 88
Non-essential		
Al	276 ± 47	421 ± 69
As	21.7 ± 4.2	21.8 ± 3.9
Cd	19.7 ± 4.0 ^a^	12.2 ± 1.8 ^a^
Pb	20.4 ± 3.6 ^b^	12.2 ± 5.2 ^b^

^a^ statistically different at level of *p* ≤ 0.05; ^b^ statistically different at level of *p* ≤ 0.01.

**Table 4 animals-13-00650-t004:** The levels of essential and non-essential metals (mean ± SEM) in ovarian tissues of selected age groups of free-ranging bitches.

Metal (ng/g)	Geographical Area			
	South		North	
	Age			
	≤3 Years (*n* = 12)	4–7 Years (*n* = 9)	≤3 Years (*n* = 12)	4–7 Years (*n* = 12)
Essential				
Cu	3656 ± 470 ^a^	2655 ± 419	2127 ± 292 ^a^	2810 ± 393
Fe	274,410 ± 31,770	396,322 ± 67,369	219,244 ± 23,922	334,995 ± 52,108
Mn	1484 ± 240	1097 ± 268	924 ± 160	1359 ± 282
Zn	68,499 ± 13,812	48,165 ± 6450	48,301 ± 8898	69,526 ± 13,521
Se	1599 ± 142	1362 ± 191	1315 ± 143	1398 ± 109
Toxic				
Al	339 ± 84	207 ± 25	376 ± 63	466 ± 124
As	21.6 ± 4.3	21.8 ± 8.4	19.8 ± 4.6	23.7 ± 6.5
Cd	10.7 ± 1.5 ^b^	31.9 ± 7.6 ^b,d^	11.4 ± 2.6	13.0 ± 2.6 ^d^
Pb	14.8 ± 4.7 ^c^	27.9 ± 4.8 ^c,e^	7.82 ± 0.73	16.5 ± 10.6 ^e^

^a,c^ statistically different at *p* ≤ 0.05; ^b,d,e^ statistically different at *p* ≤ 0.01.

## Data Availability

Data that were analyzed or generated during this study are included in this manuscript.

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
