# Peer review of "Heavy Metal(loid) Accumulation in the Ovarian Tissue of Free-Ranging Queens and Bitches Inhabiting Highly Polluted Urban Environments"

_animals, 2023, doi:10.3390/ani13040650_

Round 1

Reviewer 1 Report (Previous Reviewer 1)

The authors have now addressed all the corrections I mentioned in my previous review report (before the resubmission), especially regarding the nomenclature and methodology. Therefore, I have nothing else to add.

Reviewer 2 Report (Previous Reviewer 3)

I have no further comments to the authors and manuscript, and my doubts have been clarified.

Reviewer 3 Report (Previous Reviewer 2)

Thank you for amendments made to this article. I thoroughly enjoyed reading this submission and the topic is certainly one that needs continuing. 

This manuscript is a resubmission of an earlier submission. The following is a list of the peer review reports and author responses from that submission.

Round 1

Reviewer 1 Report

This manuscript presents interesting results regarding a biomonitoring study using bitches and queens ovaries. From my perspective, this subject has still a lot to explore and it is very relevant under a One Health perspective. Authors should receive credit for working and presenting these relevant results.

However, I have some corrections that I believe are essential to be made before publication.

1) There are some mistakes at the nomenclature used. For instance, Arsenic is not a metal, is a metaloid, so it is incorrect to include in a category named "toxic metals" (TM) or "non essential metals". You need to correct this in the whole manucsript, including the abstract. I advice the authors the use the term heavy/essential "metal(loid)s" to be 100% correct, and with this you avoid writing "metals and metalloids" all the time.

2) I am not a fan of using "essential metals" versus "toxic metals", because as you probably know, considering the principles of Toxicology, everything can be toxic, depending on the dose. And there are also intoxications or clinical signs associated with huge doses of essential metals, which are, therefore, toxic in those situations. I would advice the authors to use "essential" versus "non-essential" elements or metal(loid)s.

3) We mention that "TMs have adverse effects on ovarian function, even at relatively low-level exposures". I think it would be appropriate to illustrate which adverse effects in ovaries or reproductive function are you talking about in humans and also in mammals. Mainly because it would be useful to know why did you choose the ovaries instead other tissues (or even non-invasive simples) that are more used to biomonitoring heavy metal(loid)s than ovaries.

4) The Introduction lacks several definitions and context in my opinion. There are no definitions of "heavy metals" "metalloids" "Trace elements"... It is important to define the substance you are looking for. Heavy metal(loid)s are a very evident One Health problem, because they may deeply affect the health of living beings but also de ecossystems by changing soils, trophic chains, plant composition etc..... You should definitely present them as an One Health issue because what you find in the ovaries may reflect consequences not only on the beech and queen but also on the ecosystem they and their owners live in, even if they are asymptomatic. You mention One Health only in Discussion/Conclusion section. Perhaps, you should define One Health in the introduction.

4) In the methodology section, there are some missing information regarding the quality control method. Did you use reference materials? If yes, what were they? What was the limit of detection and the limit of quantification? How did you validate the results that come from the machine? Some essential information is missing.

5) Considering you are working with dogs and cats, I would expect to see further details in comparing habits, ages, feeding, a more detailed comparison of geographical areas (e.g. urban versus industrial versus rural areas). After all, you have most of this information in animal's clinical record. The other question that comes to me as a reader is why did you use the ovaries and not testicles from males as well? I do not see enough background reasons to perform a study with this study design because you do not present them. Even if considering it is the first survey and now you have a more detailed study planed to do, I do not see the directions of this new biomonitoring plan.

I have nothing further to add.

Reviewer 2 Report

Methods were clear however greater detail is required: 

·      Were only the ovaries analysed or did this also include the ovarian bursa? 

·      Were both the left and the right ovary analysed in all animals? If so, are the results then an average of the ovaries within that animal? 

·      Justification for the age categories would be of benefit as it is not clear why less than or more than 3 years old for bitches? (line 117). The sample numbers for the different age groups are then required. These are in the table but discussion in the results required. 

·      No consideration for power calculations towards the sample size or why those numbers yet this is considered a limitation. 

·      There is currently no ethical statement provided in this paper. This must be included. 

·      A statement about the selection of essential metals assessed would be of benefit, indicating why the chosen ones were selected. 

Very pleasing that a control group was considered, and metals present in plastic reagents was appreciated. This was not mentioned again though, so it remains unknown if metals were present in the plastic tubing. A sentence about this would be beneficial 

Discussion was ok but often lacked the criticality and relevance of findings. The conclusion appeared to be repetitive until the end where the one health element was discussed. Greater emphasis of this within the discussion should be included to capture the purpose of the study and the ‘sentinel’ idea presented at the start

Overall a nice study however some punctuation elements that need correcting throughout. The writing style could be improved to be clearer and a lot tighter without the filler words at the start of sentences such as “It should be point out, however, …”

·      Spelling edit required for “copper” on line 54 (currently ‘cupper’) 

·      Clarification of ‘difficulty metabolised’ at the end of line 61? Please re-write 

·      Remove apostrophe from the first ‘it’s’ on line 65. 

·      Missing wording on line 68 “because can damage”. For ease of reading, please modify. 

·      Sentence needs checking line 75 “have been has devoted” 

·      Check writing on line 105 “ad at the veterinary clinic” 

Reviewer 3 Report

The subject of this work is original and appropriate in this field, because animals are sensitive indicators of environmental pollution with toxic metals that can have a negative impact on human health and life.

The authors are the first researchers to address the problem of metal concentrations in the ovaries of companion animals and showed how easy it is to obtain samples for research as a result of sterilization of animals.

The methodology is generally correct, it is difficult to find fundamental errors and omissions.

 However, as the authors themselves admitted, the methodology and results obtained may be unreliable due to the limited number of samples taken, the lack of houshold animals in the experiments, and the lack of an assessment of the effect of metals on the morphology and physiology of the gonads in this study.